# Stakeholders’ Voices of Lung Cancer Screening in Hong Kong: Study Protocol for a Mixed Methods Study

**DOI:** 10.3390/healthcare12020142

**Published:** 2024-01-08

**Authors:** Fang Lei

**Affiliations:** School of Nursing, University of Minnesota Twin Cities, Minneapolis, MN 55455, USA; flei@umn.edu; Tel: +1-9525298713

**Keywords:** high-risk smokers, Hong Kong, lung cancer screening, perceptions

## Abstract

**Introduction:** This study aims to (1) explore physicians’ perceptions and experiences of lung cancer screening in Hong Kong, (2) evaluate physicians’ readiness of implementing lung cancer screening in Hong Kong, (3) explore high-risk smokers’ health beliefs of lung cancer and screening, (4) identify barriers and facilitators for high-risk Hong Kong smokers to screening for lung cancer, and (5) validate the Chinese Lung Cancer Screening Health Belief Scale in relation to high-risk smokers in Hong Kong. **Methods and analysis:** A mixed methods design will be used in this study. Individual qualitative interviews will be conducted with physicians who have experience with high-risk smokers. Physicians’ perceptions and experiences of lung cancer screening, and their readiness to accept lung cancer screening in Hong Kong, will be gathered through the qualitative interviews. A semi-structured interview guide will be used in the qualitative interviews. In addition, a quantitative survey with qualitative questions will be conducted on high-risk smokers to investigate their health beliefs of lung cancer and screening and barriers and facilitators for them to screening lung cancer. A lung cancer screening health belief scale, sociodemographic questionnaire, smoking and lung cancer screening history questionnaire, lung cancer and screening knowledge questionnaire, lung cancer stigma scale, generalized anxiety disorder scale, patient health questionnaire-9, patients’ medical trust scale and preferred lung cancer screening intervention delivery questionnaire will be conducted in the quantitative survey. Constant comparison and content analysis will be used to analyze the qualitative data. Descriptive data analysis, validity and reliability analysis, one-way analysis of variance and post hoc analyses will be used to analyze quantitative data. **Discussions:** This study explores physicians’ and high-risk smokers’ perceptions and experiences toward lung cancer screening in Hong Kong. Findings from this study can help healthcare providers and policy makers become aware of the stakeholder’s voices. In addition, these findings can help to inform the design of future interventional lung cancer screening programs and provide a tool to measure Chinese high-risk smokers’ health beliefs toward lung cancer screening. A major limitation of this mixed methods study is the amount of time taken to complete the overall study. Also, its complexity requires more collaboration and networking among researchers. **Ethics and dissemination:** This study has minimal risk to the participants. It will be submitted to the university IRB for ethical approval. Findings related to physicians’ perceptions and experiences of lung cancer screening in Hong Kong, physicians’ readiness of implementing lung cancer screening, high-risk smokers’ health beliefs of lung cancer and screening, barriers, and facilitators for high-risk Hong Kong smokers to screening lung cancer will be disseminated in journals and conferences. The reliability and validity of the Chinese lung cancer screening health belief scale will be reported in methodological research journals.

## 1. Introduction

Lung cancer is the leading cause of cancer-related deaths in the world. It has the highest mortality rates both in males and females, and according to the International Agency for Research on Cancer, lung cancer leads to an estimated 1.8 million deaths in 2020 [1]. In Hong Kong, lung cancer has been the second most common cancer since 2014, after being overtaken by colorectal cancer for two consecutive years [2]. It was the most common cancer in males and the third most common cancer in females [2]. With 3252 cases in men and 2170 cases in women, a total of 5422 new lung cancer cases were recorded in 2020, which accounted for 15.9% of all newly diagnosed cancer cases [3]. The mortality rate of lung cancer is also high in Hong Kong. It was the leading cause of cancer death both in males and in females. In 2021, about 4037 deaths from lung cancer were recorded, constituting 26.7% of all registered cancer deaths [3].

Tobacco smoking, including second-hand smoke, which is classified as Group I carcinogen, is the most important risk factor of lung cancer [4]. Compared to those who have never smoked, current smokers and ever smokers were 8.43 and 5.5 times as likely to have lung cancer, respectively [5]. People who smoke 10 or fewer cigarettes per day have an average of a 5-year shorter life expectancy than never smokers, while their lung cancer risk is up to 20 times higher than never smokers [6].

Smoking cessation or quitting smoking is the most effective measure to prevent lung cancer [7]. Quitting smoking at any age is beneficial to the smokers’ health. If a smoker stopped smoking for about 10 years, the individual’s risk of lung cancer would drop by about half. When stopped smoking for about 30 and 40 years [8,9], former smokers would reduce their risk of dying from lung cancer by 90% and 97%, respectively.

As a secondary prevention method of lung cancer, low dose computed tomography (LDCT) is an effective method of screening for lung cancer. Compared to X-ray, it can detect lung cancer at an earlier stage and decrease the mortality rate of lung cancer by 20% [10]. Lung cancer is commonly diagnosed at a late stage, and while there has been remarkable progress in treatment, the 5-year survival rate of patients with advanced lung cancer (stage IV) remains poor, at only about 4.2% [11], thus an early detection and treatment of lung cancer is essential to increase patients’ survival rate.

Since 2012, lung cancer screening with LDCT has been recommended by several countries around the world. Until 2016, three countries have enacted guidelines related to lung cancer screening, including China, the United States and Canada. However, in Hong Kong, primary prevention remains the fundamental strategy to reduce the burden of lung cancer. In June 2016, after taking into consideration local epidemiology, emerging scientific evidence and local and overseas screening practices, the Cancer Expert Working Group on Cancer Prevention and Screening [12] has fine-tuned the recommendations on lung cancer screening as follows: for the general population or high-risk populations, routine screening for lung cancer with a chest X-ray or sputum cytology is not recommended, and there is insufficient evidence to recommend for or against lung cancer screening by LDCT in asymptomatic persons or for mass screening.

This guideline is not consistent with those in the United States or in China (Table 1). According to the United States Preventive Services Task Force, adults aged 50 to 80 years old who have a 20 pack-year smoking history and currently smoke or have quit within the past 15 years should screen lung cancer annually with LDCT. Screening should be discontinued once a person has not smoked for 15 years or develops a health problem that substantially limits life expectancy or the ability or willingness to have curative lung surgery. It is a Grade B recommendation [13]. Similarly, for Chinese individuals aged 50 to 75 years old, the consensus of Chinese experts is that they should screen for lung cancer annually with LDCT if they have at least one of the following risk factors: (1) at least 20 pack-years of cigarette smoking history, including currently smoking or giving up smoking for less than 15 years; (2) passive smoking; (3) a history of occupational exposure, including asbestos, beryllium, uranium, radon, etc.; (4) a history of cancer or a family history of lung cancer; and (5) a history of chronic obstructive pulmonary disease or diffuse pulmonary fibrosis [14]. More information about other countries’ policies [15,16,17] on lung cancer screening can be found in Table 1.

Although lung cancer screening with LDCT has been recommended in many countries, the Hong Kong Cancer Expert Working Group on Cancer Prevention and Screening [12] holds a neutral attitude toward LDCT lung cancer screening. They suggested a gap in the implementation of lung cancer screening with LDCT. They recommended that more research is needed to explore the local definition of increased lung cancer risk and the screening modality or protocol for population at increased risk of lung cancer in Hong Kong.

In addition, multiple challenges exist to promote lung cancer screening with LDCT in high-risk population. Previous studies indicated that lacking access to healthcare services is a big challenge for marginal populations to screen cancers [18,19,20]. In addition, health illiteracy, language barrier, limited access to health information and cultural issues were also major barriers to cancer screening utilization in Hong Kong, especially among ethnic minorities [18,19,20]. According to the 2021 population census data, 92% of the total population in Hong Kong was Chinese, while the remaining 8% population were ethnic minorities [18]. The main language spoken in Hong Kong is Cantonese, however, only 20.4% of the ethnic minorities can read and speak Cantonese [18]. The language barrier may impact their cancer screening behavior by limiting their communication effectiveness with healthcare professionals. Although studies have been conducted to explore several other kinds of barriers to cancer screening such as breast cancer screening and colorectal cancer screening, lung cancer screening’s barriers have not been explored and were not fully understood in Hong Kong. Thus, we propose this mixed methods study, hoping to fill the gap and find the barriers and facilitators for Hong Kong high-risk smokers to screening lung cancer.

Furthermore, relevant research on exploring Chinese long-term smokers’ health beliefs of lung cancer screening has been reported. As a result, the Chinese version of lung cancer screening health belief scale has been developed to measure Chinese high-risk population’s health beliefs of lung cancer screening [21]. However, health beliefs of lung cancer screening among Hong Kong physicians and high-risk smokers have not been explored. Since the social contexts of Hong Kong and China are quite different, health beliefs of lung cancer screening could be quite different in Hong Kong than those in China. Key issues being addressed in this study are the necessity and readiness of the implementation of lung cancer screening with LDCT in Hong Kong given the vague attitude of the health administrative organization. Stakeholders’ perceptions and experiences will be explored in this study to understand their attitude toward lung cancer screening with LDCT. Barriers and facilitators for high-risk Hong Kong smokers to screening lung cancer will also be identified to help to understand the work which needs to be conducted if the recommendation will be given.

In summary, this study aims to (1) explore physicians’ perceptions and experiences of lung cancer screening in Hong Kong, (2) evaluate Hong Kong physicians’ readiness of implementing lung cancer screening, (3) explore high-risk smokers’ health beliefs of lung cancer and screening, (4) identify barriers and facilitators for high-risk Hong Kong smokers to screening lung cancer, and (5) validate the Chinese lung cancer screening health belief scale in Hong Kong high-risk smokers. These important issues need to be addressed because understanding physicians and high-risk smokers’ attitudes toward lung cancer screening is important for us to understand stakeholders’ perceptions about lung cancer screening. Disseminating information from the lung cancer screening research project will help high-risk smokers to understand their risk of lung cancer and prompt them to think about screening for lung cancer. An early detection of lung cancer can help to decrease the mortality rate of lung cancer and increase the survival rate.

## 2. Methods and Analysis

### 2.1. Research Questions

The research questions which aimed to be answered in this study are: 

1. What are physicians’ perceptions and experiences of lung cancer screening in Hong Kong? 

2. What is physicians’ readiness for implementing lung cancer screening in Hong Kong? 

3. What are high-risk smokers’ health beliefs toward lung cancer and screening? 

4. What are the barriers and facilitators for high-risk Hong Kong smokers to screening lung cancer?

### 2.2. Design

A mixed methods design will be used to explore the study aims. First, individual qualitative interviews will be used to explore physicians’ perceptions and experiences of lung cancer screening and evaluate their readiness for implementing lung cancer screening in Hong Kong. As a result of the fact that limited information about Hong Kong physicians’ perceptions and experiences of lung cancer screening could be found in literature, when such information is scarce, a qualitative design is suitable to be conducted to explore that information. In addition, the quantitative survey will be used to explore high-risk smokers’ health beliefs regarding lung cancer and screening, identify barriers and facilitators for high-risk Hong Kong smokers to screening lung cancer and validate the Chinese lung cancer screening health belief scale in Hong Kong high-risk smokers. The quantitative method is appropriate to be used for these purposes because the Chinese lung cancer screening health belief scale provided a feasible tool to explore the study aims. Using a cross-sectional survey design is efficient and cost-effective to collect data and meet the study aims.

### 2.3. Theoretical Framework

The health belief model is a social psychological health behavior change model developed to explain and predict health-related behaviors and provides a theoretical framework for the study (Figure 1) [22]. Key concepts of the health belief model are perceived susceptibility, perceived severity, perceived benefits, perceived barriers, self-efficacy and cues to action [22]. The health belief model has been used in a wide variety of health- and behavior-related studies such as cancer screening, early detection of diseases and immunizations [23].

### 2.4. Sample

A combination of convenience and chain referral sampling methods will be used to recruit participants. Inclusion criterion for the physician participants is Hong Kong physicians who have worked with high-risk Hong Kong smokers in their previous work experience. Inclusion criteria for the survey participants are people who reside in Hong Kong, are 50 to 80 years old, current smokers or quit smoking in the past 15 years and have a smoking history of at least 20 pack-year (high-risk smokers for lung cancer). Exclusion criterion for the survey participants are people without a lung cancer history or other severe diseases which shorten participants’ life expectancy. The sample size of the physician participants is determined by the data saturation. The targeted sample size of the survey participants is 270, which is a sufficient sample size calculated by the GPower software 3.1.9.6 to meet the study aims [24,25]. To detect a medium effect (d = 0.5), alpha error probability equals to 0.05 and power equals to 0.95. Using an independent sample t-test to detect differences will require a total sample size larger than 210 in the formal study. Taking the attrition and response rate into consideration, 270 would be a sufficient sample size for the survey.

### 2.5. Setting and Recruitment Procedures

The physicians will be recruited with the help of the Hong Kong Medical Association. The Hong Kong Medical Association was founded in 1920 and has more than 8000 members from all sectors of medical practice. Close contact has been established with the association. Online recruitment flyers will be distributed to the association’s members by emails, and any member who is willing to take part in the study will be asked to contact the primary investigator (PI) of this study through email. A follow up screening call/email will be made/sent to the potential physician participant. If the physician is eligible to take part in the study, the information sheet about the study will be sent to the physician participant and a telephone interview appointment will be made with the physician. After the interview, a gift card with 300 HKD value will be emailed to the physician for compensating their time in the study.

To recruit high-risk smokers to the study, recruitment flyers will be distributed in communities in Hong Kong with the help of the community office staffs. In addition, the recruitment flyers will be posted in the bulletin board of the communities. The PI’s contact information—including telephone number, email address, and a scan code with the PI’s WhatsApp contact number—will be provided in the recruitment flyers. To ensure the success of the recruitment, an alternative recruitment method is prepared for which we are not able to recruit sufficient participants into the study. We will ask the contacted physician participants to recommend eligible high-risk smoker participants into the study to expand the participant pool. When a potential participant contacts the PI, the PI will screen him/her for the eligibility to take part in the study. If the participant is eligible to take part in the study, the information sheet about the study will be sent to the participant and the e-version questionnaires will be sent to the participants by email once the participant consents to take part in the study. Reminder emails will be sent to the participants weekly to remind them to complete the questionnaires. Once the participant returns their responses to the questionnaires, a gift card with 100 HKD value will be emailed to the participant as compensation for their time in the study.

### 2.6. Instruments

**Semi-structured individual interview guide.** A semi-structured interview guide (Table 2) was developed based on the evidence from the relevant literature. Physicians’ knowledge about lung cancer screening, attitude toward lung cancer screening with low dose computed tomography and readiness to implement lung cancer screening will be asked in the interviews.

**Sociodemographic Questionnaire.** High-risk Hong Kong smokers’ sociodemographic characteristics will be measured by the sociodemographic information questionnaire. The questionnaire includes 10 questions asking about participants’ age, gender, marital status, number of children, level of education, income, religion, occupation, health insurance status and the total years lived in Hong Kong.

**Health Beliefs of Lung Cancer Screening Scale**. High-risk Hong Kong smokers’ health beliefs of lung cancer screening will be measured by the Chinese lung cancer screening health belief scale, a scale which was generated from our preliminary studies. The Chinese Lung Cancer Screening Heath Belief Scale was adapted from the English version of the lung cancer screening health belief scale [26]. The original scale was translated using the Brislin’s back-translation approach [27]. All the translation team members were fluent in both English and Chinese. The instrument was reviewed by five professional experts in cancer nursing and cross-cultural research, and five participants by cognitive individual interviews. The instrument has been validated and proven to be reliable in Chinese Americans. It included 57 items and 6 subscales, of which the content was proven to be highly valid through the expert review and participants’ review, with the item-level content validity index ranging from 0.8 to 1 and the subscale-level content validity index/universal agreement ranging from 0.75 to 1.

**Smoking and Lung Cancer Screening History Questionnaire.** Information on participants’ smoking and lung cancer screening history (current smoking status, histories of smoking and lung cancer screening and family history of lung cancer) will be collected through the smoking and lung cancer screening history questionnaire.

**Lung Cancer and Screening Knowledge Questionnaire.** Participants’ lung cancer and screening knowledge will be asked using 12 questions from the lung cancer and screening knowledge questionnaire [28]. Internal consistency reliability of the scale was acceptable (0.66) and test-retest reliability of the overall scale was 0.84 (intraclass correlation).

**Shortened Version-Cataldo Lung Cancer Stigma Scale.** Twenty-one questions will be asked about participants’ lung cancer stigma by the shortened version-Cataldo lung cancer stigma scale using four-point Likert-style responses. Reliability scores of the scale range from 0.75 to 0.96, which are in the acceptable to excellent range [29].

**Generalized Anxiety Disorder Scale.** Seven questions will be asked about participants’ anxiety level through the generalized anxiety disorder scale using four-point Likert-style responses. The internal consistency of the generalized anxiety disorder scale is 0.92, which is at the excellent level [30].

**Patient Health Questionnaire-9.** Nine questions will be asked about participants’ depression level by patient health questionnaire-9 using four-point Likert-style responses. The internal consistency of the patient health questionnaire-9 is 0.89, which is at the excellent level [31].

**Patients’ Medical Trust Scale.** Five questions will be asked about participants’ medical trust by patients’ medical trust scale using 5-point Likert-style responses. The internal consistency of the patients’ medical trust scale is 0.77, which is at the acceptable level [32].

**Preferred Lung Cancer Screening Intervention Delivery Questionnaire.** The preferred lung cancer screening intervention delivery questionnaire was developed by the research team. Participants’ preferred intervention delivery methods among the choices of remote (online internet-based workshop, phone consultation, text message information delivery, interactive app delivery) vs. in-person interventions (focus group vs. individual one-on-one vs. couple-based face-to-face interventions) and self-learning booklets vs. phone follow-up inquiry will be asked using 10 questions from the questionnaire. This questionnaire is an assessment questionnaire, and results could be used in the later interventional lung cancer screening program.

### 2.7. Data Analysis

The qualitative data will be analyzed using constant comparison and thematic coding methods. Significant themes will be summarized and extracted to reflect physicians’ perceptions and experiences of lung cancer screening in Hong Kong. Physicians’ readiness of implementing lung cancer screening in Hong Kong will also be identified in the qualitative data analysis. The coding will be conducted line-by-line using NVivo 14.0 software. The open coding method will be used for the initial coding. Constant comparison between codes and thematic coding will be utilized to categorize significant themes together.

The quantitative data will be analyzed by the PI with the help of a statistician using the SPSS 27.0 software. One-way analysis of variance and post hoc analyses will be used to identify facilitators and barriers to lung cancer screening in high-risk Hong Kong smokers. Descriptive data analysis (mean and standard deviation, frequency and percentage) and independent samples *t*-test will be used to explore high-risk smokers’ health beliefs of lung cancer and screening. *p* is set at the 0.05 level. If the *p* value is less than 0.05, the result is significant. The validity and reliability of the Chinese lung cancer screening health belief scale will be tested using construct validity, criterion-related validity and internal consistency reliability. The exploratory and confirmatory factor analysis, the known group comparison approach and Cronbach’s alpha will be used [33].

### 2.8. Patient and Public Involvement

Research questions related to this study were developed based on the literature review and clinical practice dilemma. The first time when the public is involved in the research is when they are recruited to take part in the study. They will be screened for their eligibility for participating in the study and provided information. Findings from this study could further benefit high-risk members of public. Involvement of the public in the conduct, recruitment and outcome dissemination of this study ensures the study’s feasibility, applicability and extendibility. Implementation of this study could raise high-risk smokers’ awareness of lung cancer screening and provide evidence for future interventional screening projects, which could eventually lead to an increased uptake rate of lung cancer screening and decreased mortality rate of lung cancer in members of the public who are considered high-risk.

## 3. Discussion

### 3.1. Strengths

This study explores physicians’ and high-risk smokers’ perceptions and experiences toward lung cancer screening in Hong Kong. Knowledge about barriers and facilitators for high-risk Hong Kong smokers to screening lung cancer will also be explored. Findings from this study can help healthcare providers, the public and policy makers have an overview of the status of lung cancer screening practice and understand the barriers and facilitators for the high-risk smokers to screening lung cancer.

These findings can help healthcare providers and policy makers become aware of the voices of the stakeholders. In addition, these findings can help to inform the design of future interventional lung cancer screening programs and provide a tool to measure Chinese health beliefs toward lung cancer screening. It may help to increase the uptake of lung cancer screening among high-risk smokers and provide recommendations to the lung cancer screening policy in Hong Kong.

In the short term, heavy smokers in Hong Kong will benefit from this study. This research program will serve as a vehicle to disseminate information on lung cancer screening in high-risk Hong Kong smokers. Getting information on lung cancer screening with LDCT can help high-risk smokers understand their risk of lung cancer and the detection method for benefiting them from detecting lung cancer earlier. It will help to increase high-risk smokers’ awareness about lung cancer and screening, and therefore may trigger them to take active action to detect lung cancer. In the medium and long term, this study will provide evidence to the designing of future lung cancer screening programs. Barriers to lung cancer screening will be addressed in the interventions. Facilitators to lung cancer screening will be utilized to increase high-risk smokers’ uptake of lung cancer screening. Interventions on lung cancer screening will eventually help to decrease the mortality rate of lung cancer (which can be measured by the number of deaths caused by lung cancer scaled to the size of the overall population) and increase lung cancer survival rate (which can be measured by the portion of people with lung cancer diagnose that will be alive after a given time range). It will have significant influence on lung cancer screening policy and help to decrease the medical burden on the public health system.

### 3.2. Limitations

A major limitation of this mixed methods study is the amount of time taken to complete the overall study. Also, its complexity requires more collaboration and networking among researchers. In addition, the cost to complete a mixed methods study is usually higher than that of a qualitative or quantitative study alone. However, with both the quantitative and qualitative research experience gained in previous studies and the close collaboration with local physician organizations and research teams, we are positive that our study can proceed successfully as planned.

### 3.3. Dissemination

After the study is completed, findings related to physicians’ perceptions and experiences of lung cancer screening in Hong Kong, physicians’ readiness of implementing lung cancer screening, high-risk smokers’ health beliefs regarding lung cancer and screening, barriers, and facilitators for high-risk Hong Kong smokers to screening lung cancer will be disseminated in journals and conferences. The reliability and validity of the Chinese lung cancer screening health belief scale will also be reported in methodological research journals. In addition, the research findings will be disseminated and shared with public and health administrative organizations. This may raise the discussion about lung cancer screening and bring a shift on lung cancer screening policy, which may change the practice and benefit the high-risk population of lung cancer.

## Figures and Tables

**Figure 1 healthcare-12-00142-f001:**
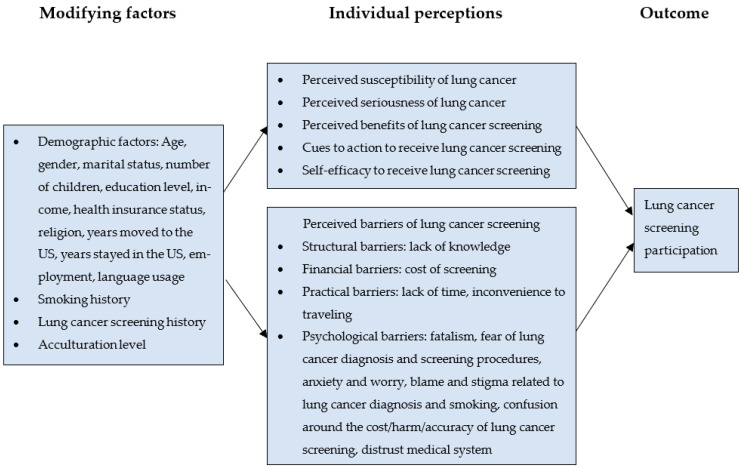
Theoretical framework for lung cancer screening participation.

**Table 1 healthcare-12-00142-t001:** Guidelines for screening lung cancer in different countries/areas.

Country/Area	Guideline
United States	Adults aged 50 to 80 years old who have a 20 pack-year smoking history and currently smoke or have quit within the past 15 years should screen lung cancer annually with LDCT. Screening should be discontinued once a person has not smoked for 15 years or develops a health problem that substantially limits life expectancy or the ability or willingness to have curative lung surgery.
China	Chinese individuals aged 50 to 75 years old, with at least one of the following risk factors, should screen lung cancer annually with LDCT: (1) at least 20 pack-years of cigarette smoking history, including currently smoking or giving up smoking for less than 15 years; (2) passive smoking; (3) a history of occupational exposure, including asbestos, beryllium, uranium, radon, etc.; (4) a history of cancer or a family history of lung cancer; and (5) a history of chronic obstructive pulmonary disease or diffuse pulmonary fibrosis.
Hong Kong	For the general population or high-risk populations, routine screening for lung cancer with a chest X-ray or sputum cytology is not recommended, and there is insufficient evidence to recommend for or against lung cancer screening by LDCT in asymptomatic persons or for mass screening.
Canada	For adults aged 55 to 74 years with at least a 30 pack-year* smoking history who currently smoke or quit less than 15 years ago, annual screening with LDCT up to three consecutive times is recommended. Screening should only be carried out in healthcare settings with expertise in early diagnosis and treatment of lung cancer (weak recommendation with low quality evidence) [15].
European Union	A validated risk stratification approach should be used to select people who should be screened for future lung cancer low-dose CT programs [16].
Japan	For people who are aged 50 or over with a Brinkman index ≥600 (i.e., ≥30 pack-years of smoking), LDCT screening may be considered for population-based screening [17].

**Table 2 healthcare-12-00142-t002:** Semi-structured Interview Guide.

Thank you for your interest in this study. I am going to ask you some questions about your knowledge of lung cancer screening, attitude toward lung cancer screening with low dose computed tomography and readiness to implement lung cancer screening. Your responses to the questions will be recorded by the digital recorders. Our conversation is confidential and no information about your identity will be shared.
**Knowledge about lung cancer screening**
1. How much do you know about lung cancer screening?
2. Who do you think should screen lung cancer annually?
3. What method do you think should be used for screening lung cancer?
4. What benefits and risks do you think may be associated with lung cancer screening?
**Attitude toward lung cancer screening with low dose computed tomography**
1. What is your attitude toward lung cancer screening with low dose computed tomography?
2. Did you recommend your patients who have a high risk for lung cancer to screening for lung cancer? What are the reasons for your recommendation/not having such recommendation?
3. Do you support screening lung cancer with low dose computed tomography among people who have high risk for lung cancer? Why?
4. Do you think that both public and private insurances should cover the cost of lung cancer screening in high-risk population? Why?
**Readiness to implement lung cancer screening**
1. Do you think you are ready to recommend lung cancer screening with low dose computed tomography to your patients who have high risk of lung cancer?
(Prompt): If yes, what efforts have you put into that enables you to do so? If not, what help do you think is needed for you to do so?
2. Do you think a guideline about screening for lung cancer is needed to be enacted in Hong Kong?
(Prompt): If yes, who do you think should make efforts to enable that guideline to be enacted? If no, what are the reasons? When do you think is a good time to enact that guideline?
3. What works need to be done before that guideline can be enacted?

## Data Availability

Any data generated from this study will be available to be shared upon request submitted to the primary investigator of this study.

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
