# Peer review of "Stakeholders’ Voices of Lung Cancer Screening in Hong Kong: Study Protocol for a Mixed Methods Study"

_healthcare, 2024, doi:10.3390/healthcare12020142_

Round 1
Reviewer 1 Report
Comments and Suggestions for Authors
Strengths: This is a great addition to the field. This study explores physicians’ and high-risk smokers’ perceptions and experiences toward lung cancer screening in Hong Kong.
Study method: It is impressive. There is a wonderful sample size, and the protocols are clear. You may want to be careful in terms of stating it is the first study, maybe the "is the first known study".
Intro:
I would suggest that you increase the number of references in the Introduction, as it seems a bit sparce in evidence of your statements. Add additional references supporting those like [13].
Consider add a visual that details the differences between protocols in reference to "This guideline is not consistent with those in the United States or in China." A simple summary table maybe.
There are a ton of research questions: consider adding them together or number them: "The research questions which aimed to be answered in this study are: What are physicians’ perceptions and experiences of lung cancer screening in Hong Kong? What is physicians’ readiness of implementing lung cancer screening in Hong Kong? What are high-risk smokers’ health beliefs toward lung cancer and screening? What are the barriers and facilitators for high-risk Hong Kong smokers to screening lung cancer?" As is, it is a bit too much.
Methods:
Expand the Theoretical Framework with more details (2-4 more sentences and references)
Results: Be careful to make the claim that this is the first study on this topic. It is in combination but there are others that you should reference like: Abdullah, A. S. M., Rahman, M. A., Suen, C. W., Wing, L. S., Ling, L. W., Mei, L. Y., ... & Kwan, Y. H. (2006). Investigation of Hong Kong doctors' current knowledge, beliefs, attitudes, confidence and practices: implications for the treatment of tobacco dependency. Journal of the Chinese Medical Association, 69(10), 461-471.
Overall: very well done. Look forward to the results of the study in another publication.
Comments on the Quality of English Language
Some minor editing
Reviewer 2 Report
Comments and Suggestions for Authors
Presentation of the Research Protocol to know the stakeholders of lung cancer screening in Hong Kong.
The study presents 5 main objectives to be achieved through mixed methods (qualitative/quantitative):
1. Explore physicians´ perceptions and experiences of lung cancer screening (LCS)
2. Evaluate physicians´ readiness to implement LCS
3. Explore high-risk smokers´ health beliefs of LCS
4. Identify barriers and facilitators of high-risk smokers.
5. Validate the Chinese LCS Health Belief Scale for high-risk smokers.
The protocol includes the ethical aspects and its approval by an IRB. It also mentions how the results will be disseminated.
The introduction and referenced bibliography are complete and up to date. Minor Topics:
1. Incidence and mortality data are local. Global statistics from IARC can also be used.
2. Randomized clinical trials that have demonstrated the efficacy of LDCT screening (American and European) are not referenced.
3. There are more recommendations from around the world on the LCS. Can be reviewed.
Methods and Analysis
Regarding the design of the qualitative study, it was carried out individually. For the objectives of the study, it is more convenient to explore physicians’ perceptions and experiences of working in discussion groups. For the research with high-risk smokers, you can also consider adding a discussion group methodology.
Regarding the theoretical framework, Figure 1 does not appear in the document.
Regarding the sample, to be representative of the population, it must be segmented into different variables to avoid bias. For example: age, sex, socio-economic level, or place of residence... The sample will depend on the segmentation of the sample. There is no reference to the maximum statistical error associated with the sample, type of randomization, or confidence level. In the sample, it is also appropriate to list the exclusion criteria.
Regarding the recruitment procedures, it is very generic. Financial remuneration will depend on the approval of the ethics committee.
Reviewer 3 Report
Comments and Suggestions for Authors
Although, the topic is interesting, I have several methodological concerns:
Lines 45-41: The data (2014 and 2015) are too old. The authors should provide the latest available information.
Lines 104-106: The authors should provide more details with respect to the barriers.
Lines 112-121: The authors should rewrite the paragraph providing more information.
Lines 122-126: The research questions should be mentioned in the “Methods and Analysis” Section.
Lines 135-141: The objective of the study should be mentioned in the “Introduction” Section.
Lines 156-159: The authors should expand the paragraph providing more information.
Lines 160-171: The authors should provide detailed information with respect to sample size calculation.
Lines 229-260: The authors should provide more details with respect to the questions.
Lines 261-274: The authors should provide more details with respect to the data analysis.
Lines 286-315: The “Discussion” is too short.
Round 2
Reviewer 2 Report
Comments and Suggestions for Authors
The authors have reviewed most of the recommendations made by the reviewers. They have significantly improved the article. We understand that some of the recommendations on the design of the study have not been introduced due to their technical difficulty (sample segmentation and exclusion criteria).
We wish you the best of luck, energy, and time to carry out this research protocol.
Reviewer 3 Report
Comments and Suggestions for Authors
Thank you for responding to the comments.
However, in line 295, the letter "P" should be replaced by the letter "α" (significance level).